# A pre-training and self-training approach for biomedical named entity recognition

**Shang Gao**[ID][1]*, **Olivera Kotevska**[2], **Alexandre Sorokine**[ID][3], **J. Blair Christian**[ID][1]*

**1** Computational Sciences and Engineering Division, Oak Ridge National Laboratory, Oak Ridge, TN, United States of America, **2** Computer Science and Mathematics Division, Oak Ridge National Laboratory, Oak Ridge, TN, United States of America, **3** Geospatial Science and Human Security Division, Oak Ridge National Laboratory, Oak Ridge, TN, United States of America

* gaos@ornl.gov (SG); christianjb@ornl.gov (JBC)

**Data Availability Statement:** All data used for this experiment are openly available online. Semantic Medline can be downloaded from https://skr3.nlm.nih.gov/SemMedDB/. MedMentions can be downloaded from https://github.com/chanzuckerberg/MedMentions. The NER datasets

## Abstract

Named entity recognition (NER) is a key component of many scientific literature mining tasks, such as information retrieval, information extraction, and question answering; however, many modern approaches require large amounts of labeled training data in order to be effective. This severely limits the effectiveness of NER models in applications where expert annotations are difficult and expensive to obtain. In this work, we explore the effectiveness of transfer learning and semi-supervised self-training to improve the performance of NER models in biomedical settings with very limited labeled data (250-2000 labeled samples). We first pre-train a BiLSTM-CRF and a BERT model on a very large general biomedical NER corpus such as MedMentions or Semantic Medline, and then we fine-tune the model on a more specific target NER task that has very limited training data; finally, we apply semi-supervised self-training using unlabeled data to further boost model performance. We show that in NER tasks that focus on common biomedical entity types such as those in the Unified Medical Language System (UMLS), combining transfer learning with self-training enables a NER model such as a BiLSTM-CRF or BERT to obtain similar performance with the same model trained on 3x-8x the amount of labeled data. We further show that our approach can also boost performance in a low-resource application where entities types are more rare and not specifically covered in UMLS.

## Introduction

Named entity recognition (NER) is a critical component for many downstream applications, such as information retrieval, information extraction, and question answering. NER is especially important in the domain of biomedical literature mining, where it is becoming more difficult for individuals to keep up with the sheer volume of new research being published. Building effective NER approaches that can effectively identify biomedical concepts such as diseases, chemicals, and proteins can aid researchers in finding and identifying relevant research and speed up the process of scientific discovery.

can be downloaded from the BioBert GitHub page at https://github.com/dmis-lab/biobert. The TAC SRIE dataset can be downloaded from https://tac.nist.gov/2018/SRIE/data.html.

**Funding:** Blair Christian (BC) at Oak Ridge National Laboratory received funding from the Department of Energy (energy.gov). This funding was provided through the Laboratory Directed Research and Development (LDRD) program of Oak Ridge National Laboratory, under LDRD project No. 9494. These funds were used to facilitate this study and support of salaries for SG, OK, AS, and BC. This manuscript has been authored by UT-Battelle, LLC under Contract No. DE-AC05-00OR22725 with the U.S. Department of Energy. The United States Government retains and the publisher, by accepting the article for publication, acknowledges that the United States Government retains a non-exclusive, paid-up, irrevocable, world-wide license to publish or reproduce the published form of the manuscript, or allow others to do so, for United States Government purposes. The Department of Energy will provide public access to these results of federally sponsored research in accordance with the DOE Public Access Plan (http://energy.gov/downloads/doe-public-access-plan). This research used resources of the Oak Ridge Leadership Computing Facility at the Oak Ridge National Laboratory, which is supported by the Office of Science of the U.S. Department of Energy under Contract No. DE-AC05-00OR22725. This research used resources of the Compute and Data Environment for Science (CADES) at the Oak Ridge National Laboratory, which is supported by the Office of Science of the U.S. Department of Energy under Contract No. DE-AC05-00OR22725. The funding offices from the Office of Science and DOE did not have any additional role in the study design, data collection and analysis, decision to publish, or preparation of the manuscript. The specific roles of all authors are articulated in the 'author contributions' section.

**Competing interests:** The authors have declared that no competing interests exist.

Existing NER tools can be broken down into two broad categories: rule-based methods and machine learning methods [1–4]. Rule-based methods require human experts to manually hand-craft specific rules to identify different types of named entities—examples include term-matching with an existing concept database such as the Unified Medical Language System (UMLS) [5, 6] or pattern matching based on part-of-speech and sentence structure [7, 8]. In practice, rule-based methods require expensive expert knowledge to develop and tend to work only within a very limited domain on which the rules were developed. Furthermore, in the domain of biomedical literature, rule-based approaches often fail to adapt to novel concepts and vocabulary that are characteristic of new scientific publications.

On the other hand, machine learning approaches automatically learn patterns for identifying named entities using a large corpus of labeled training data. In general, machine learning approaches tend to be more flexible than rule-based approaches; however, they require large volumes of word-level annotations which are expensive and difficult to obtain in biomedical settings [9]. The generalizability and accuracy of machine learning approaches, especially in the case of newer deep learning models, are heavily dependent on the amount of labeled data available. In biomedical NER, annotated data is often limited to only a particular type of entity such as chemicals or genes; as a result, existing machine learning NER tools can be limited in scope in that they can only identify a very limited set of entity types.

Developing effective biomedical NER systems for a new application area can be difficult if there is very limited annotated training data, as obtaining gold standard biomedical annotations often requires expensive expert knowledge. In this work, we address this challenge by utilizing a combination of transfer learning, in which we first pre-train a model using a large annotated NER corpus from an adjacent domain, and semi-supervised learning, in which we generate pseudo-labels on unlabeled data from the target domain to improve the performance of our NER model. Using a base NER model such as the popular Bidirectional Long Short Term Memory Conditional Random Field (BiLSTM-CRF) [10] or state-of-the-art Bidirectional Encoder Representations from Transformers (BERT) [11], we show that the combination of transfer learning and semi-supervised learning can significantly reduce the amount of labeled data required to obtain strong performance. Our method obtains F1 scores comparable to a fully supervised model trained on 3x-8x the amount of labeled data when evaluated on eight standard biomedical NER benchmarks. To our knowledge, there does not exist any previous work that thoroughly examines the cumulative effect of transfer learning and semi-supervised learning in the biomedical NER space. Our contributions are as follows:

- We evaluate the benefits of pre-training on three different corpora for biomedical named entity recognition using two common NER approaches—BiLSTM-CRF and BERT—and eight standard biomedical NER datasets covering common biomedical entity types such as chemicals, genes, and diseases.

- We explore the benefits of semi-supervised self-training with different amounts of labeled data using the BiLSTM-CRF and BERT on the same eight standard biomedical NER datasets.

- We show that by combining transfer learning with self-training, a NER model such as a BiLSTM-CRF or BERT can obtain similar performance to a fully supervised model while using only 12%-30% of the total available training data.

- We show that semi-supervised self-training can propagate errors and lower the F1 score when initial model performance is low, and that transfer learning can be critical in low data settings (250-500 labeled samples) to get the initial model performance to a level where semi-supervised learning can be effective.

- We evaluate the effectiveness of pre-training on UMLS entity types and then applying self-training on a downstream NER task where the entities of interest are not the same entity types as those covered in UMLS; we show that these methods can still improve performance.

## Related work

**Methods for named entity recognition.** Traditional NER approaches generally utilized manually crafted expert rules and heuristics and to identify entities of interest [12–18], such as persons, locations, and organizations; these types of rule-based approaches are still in use today in domain areas such as medicine where labeled training data is difficult to obtain [8, 19, 20]. Recent work has shown that supervised machine learning approaches, especially deep neural networks, achieve superior performance on various NER tasks [1, 2, 21]. The BiLSTM-CRF architecture is extremely popular in NER applications due to its strong performance on a wide range of sequence tagging tasks [10, 22–24]. More recently, BERT has shown state-of-the-art performance across a wide range of natural language processing tasks including named entity recognition [11, 25, 26]. Existing NER work utilizing BiLSTM-CRF and BERT-based models often focus on supervised applications that often require tens of thousands or more manually annotated sentences. In this work, we extend these two popular approaches to biomedical NER settings with very few labeled examples by incorporating transfer learning and semi-supervised techniques.

**Transfer learning in NER.** In transfer learning, a model that is trained on one task is then retrained on and applied to a different related task; knowledge gained when training on the first task may boost performance on the second task, especially when labeled training data is scarce for the second task [27]. A common example is downloading an image classifier that is already trained on the very large ImageNet dataset and then fine-tuning it on a specific image classification task of interest—this often achieves better performance than training on the downstream task only. Transfer learning is highly effective across a wide range of different applications in image recognition and natural language processing [28–31].

In this study, we build upon previous work that explores how to effectively leverage transfer learning for biomedical NER. [32] showed that pre-training a BiLSTM-CRF on a silver-standard corpus of 50K abstracts, tagged for biomedical entities by automated tools rather than human experts, can boost performance on downstream biomedical NER tasks that have fewer than 6K training samples. Similarly, [33] showed that pre-training a BiLSTM-CRF model on a silver standard corpus of 5M sentences from PubMed abstracts, tagged using a trained CRF model rather than human experts, boosts performance on downstream biomedical NER tasks for different entity types. Other work, including [34–38], explore other variations of transfer learning and come to similar conclusions that transfer learning can significantly improve performance on downstream NER tasks. We extend these previous works by (1) comparing the effectiveness of three NER pre-training corpora of differing size and quality and (2) incorporating semi-supervised learning after transfer learning to further improve the performance of our NER approaches.

In the context of BERT, it can be argued that any application that utilizes BERT also utilizes transfer learning—BERT is pre-trained on a very large corpus of unlabeled text using masked-language-modeling or a similar pre-training task and then fine-tuned on a downstream application [11]. Several previous studies have simply taken BERT models pre-trained on different corpora and then applied them to various downstream NER tasks [39–41]. In our work, we first take a BERT model that has been pre-trained on biomedical abstracts and then further pre-train it on a NER task (as opposed to a generic language modeling task); we evaluate

if this second round of pre-training boosts performance on downstream biomedical NER applications.

**Semi-supervised learning in NER.** In semi-supervised learning, a machine learning model is trained using both labeled and unlabeled data—the model is trained using pseudo-labels or other patterns from the unlabeled data, which can provide a performance boost especially in applications where labeled data is limited [42]. There are many different types of semi-supervised learning, but a simple example is to train a classifier on the labeled data and then use it to predict on the unlabeled data—samples with high prediction confidence are assumed to be labeled correctly and used to expand the labeled training set. Like with transfer learning, semi-supervised learning has been widely and successfully applied in a range of different applications [43–46].

Several previous works [47–49] have successfully applied semi-supervised methods in the context of NER. These methods generally involve using a combination of existing predictive models, feature similarity metrics, and heuristics to generate NER pseudo-labels on an unlabeled dataset; the pseudo-labels with the highest confidence are then added to the original training data and used to train an improved NER model. In this work, we use an extremely simple semi-supervised technique—self-training—in combination with transfer learning and show that this potent combination can significantly improve the performance of NER models in biomedical settings with very few labeled training examples, especially when the entities of interest overlap with those covered in the pre-training dataset.

## Materials and methods

### Problem description and proposed solution

In this work, we address the standard NER task in which we have a corpus of text segments, typically at the sentence level, in which each text segment may contain one or more named entities. Each named entity can consist of one or more consecutive words. Given an unannotated text segment $T$ consisting of words $w_0, w_1, \ldots, w_{l-1}, w_l$ and containing a set of named entities $E = \{e_0, e_1, \ldots, e_{n-1}, e_n\}$ where each entity corresponds with one or more consecutive words, a model $M$ must correctly identify the start and end words of each named entity within $E$. A commonly used method to frame this problem is the BIO annotation scheme, in which each word $w_i$ in $T$ is tagged as either B (first word of a named entity), I (non-first word belonging to a named entity), or O (does not belong to a named entity). This annotation scheme allows for easy parsing of the positions of the entities in $E$, especially among entities that share neighboring word boundaries. Thus, the NER task can be framed as a sequence tagging task in which each word $w_i$ in $T$ is treated as a three-class classification problem.

Whereas previous works in NER sequence tagging often focus on the supervised setting in which there are thousands or tens of thousands of annotated training examples, we specifically focus on settings where (1) there are limited annotated training examples in the target domain (between 250 and 2000 labeled sentences in our case), i.e., the train dataset, (2) there is access to unannotated text segments within the same target domain, i.e., the unsupervised dataset, and (3) there exist one or more corpora of annotated training data from a neighboring or related domain, i.e., the pre-training dataset.

To address the challenges associated with limited training data within the target domain, we first pre-train a model on the annotated data from the pre-training dataset and then use the limited annotated data from the train dataset to further fine-tune the model. Finally, we apply semi-supervised learning on the remaining unannotated data in the unsupervised dataset to further boost the performance of the model. Our overall workflow is illustrated in Fig 1, and we explain each step in greater detail in the following subsections.

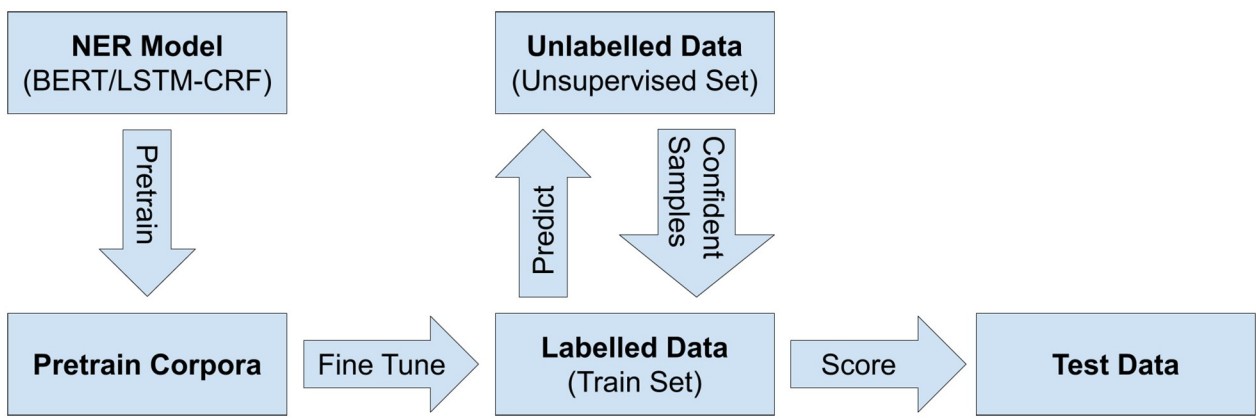

**Fig 1. NER workflow used for our experiments.**

## NER models

For our NER models, we utilize a BiLSTM-CRF, which is a widely used architecture for sequence-tagging tasks, and BERT, which is a relatively new architecture that is state-of-the-art or close to state-of-the-art in many NER tasks including biomedical NER.

For our BiLSTM-CRF model, we utilize publicly available Word2Vec embeddings of dimension size 200 that are pre-trained on PubMed and PMC texts. Because our word embeddings are trained on all of PubMed and PMC, our word embedding matrix contains approximately 2.3 million unique words [50]; however, our pre-training and NER datasets only use a small fraction of this total vocabulary. Therefore, we freeze the word embeddings during training (rather than initializing them as trainable parameters) to reduce overfitting and improve the generalizability of our BiLSTM-CRF.

Our BiLSTM-CRF model architecture consists of two bidirectional LSTM layers with 300 units each, followed by a CRF classification layer. All training is performed using the Adam optimizer with batch size 128 and learning rate $1e - 4$. We note that while recent work introduces more complex sequence tagging architectures, such as incorporating character-level inputs [51] and convolutional neural networks (CNNs) [52], we kept our BiLSTM-CRF model fairly simple to show that our approach works with both simple and state-of-the-art models.

For our BERT model, we utilize the pre-trained WordPiece vocabulary and model weights from BlueBERT Base [53], which is the BERT Base model pre-trained on PubMed abstracts and MIMIC III clinical notes, as this model has shown superior performance on biomedical and medical NER tasks compared to other BERT-based models such as BioBERT [40]. We note this version utilizes an uncased vocabulary. For additional information about the architecture of BERT, we refer readers to previous work that describes BERT thoroughly [54–58].

We utilize the standard token classification setup for BERT, in which a sequence of input tokens is processed by the BERT model, and then each output token is passed to a dense linear layer followed by a softmax classification layer that assigns labels. We note that BERT utilizes the WordPiece tokenizer that breaks up long words into subword tokens; however, all our ground truth labels for NER tagging are at the word level rather than the subword level. Following previous work on applying BERT for NER [59], during training and inference, we only use the label from the first subword token associated with each word. All models are implemented using the Huggingface Transformers library [60], and training is performed using the Adam optimizer with batch size 32 and learning rate $5e - 5$.

As a final baseline, we include the performance of two out-of-the-box tools which are popular resources for performing biomedical NER—scispaCy [61] and MetaMap [62]. ScispaCy is a deep-learning-based approach trained on the MedMentions dataset, while MetaMap is a rule-based approach that utilizes a manually curated dictionary. Because these two tools can perform NER without requiring any additional labeled training data, any method that utilizes supervised training on labeled data should at least outperform these two tools to be considered practically useful.

## Transfer learning

To alleviate the limitations associated with a small number of labeled examples, we evaluate the effects of transfer learning in which we first pre-train our models on a large NER dataset from a related domain and then fine-tune the model weights on the target NER dataset. For our pre-training datasets, we utilize Semantic Medline (available online at url https://skr3.nlm.nih.gov/SemMedDB/), which consists of approximately 28M PubMed abstracts that are automatically annotated for all UMLS entities using the rule/dictionary-based MetaMap tool [63, 64], and MedMentions (available online at url https://github.com/chanzuckerberg/MedMentions), which consists of approximately 4K abstracts manually annotated for UMLS entities by human experts [65]. For our pre-training datasets, we utilize sentence-level inputs annotated using word-level BIO labels without entity type. We generate three different pre-training datasets—∼100K annotated sentences randomly sampled from Semantic Medline, ∼1M annotated sentences randomly sampled from Semantic Medline, and all ∼50K sentences from the MedMentions dataset. Detailed dataset descriptions are available in Table 1.

For each of the pre-training datasets, we use 80/20 splitting to create training and validation sets. For the BiLSTM-CRF, we train on the training set and validate on the validation set after each epoch, stopping training when validation exact-F1 stops improving for five consecutive epochs. For BlueBERT, we use the same setup for the MedMentions dataset; however, we observed that using this setup on Semantic Medline dataset causes BlueBERT to overfit and significantly reduces performance on downstream tasks—this is likely because (1) the BERT model has 340M learnable parameters and can learn extremely nuanced patterns and (2) the labels in Semantic Medline are more prone to errors because they are annotated by a rule-based method. Therefore, we limit the training to a single epoch on both Semantic Medline datasets.

## Supervised fine-tuning

Once the model has been pre-trained on one of the pre-training datasets, we fine-tune it using the target NER dataset. In our experimental setup, we assume that only a fraction of sentences within the target NER dataset has annotations. For example, in a dataset with 10K total sentences, only 500 sentences may have gold-standard annotations.

We use 80/20 splitting on the annotated subset of the dataset to create a train and validation set. We initialize the model using the weights obtained from the pre-training step, and then train on the train set, validating on the validation set after every epoch. Training stops when validation exact-F1 stops improving for ten consecutive epochs.

**Table 1. Detailed information about each of our pre-training datasets.**

| Name | Entity Types | Num Sentences | Num Entities | Entity Words/Total Words |
|------|-------------|---------------|--------------|--------------------------|
| SemMed 100K | All UMLS, Rule-Based Annotations | 95,607 | 234,807 | .2696 |
| SemMed 1M | All UMLS, Rule-Based Annotations | 953,589 | 2,284,983 | .2729 |
| MedMentions | All UMLS, Human Expert Annotations | 47,722 | 321,899 | .4138 |

## Semi-supervised learning

We use a simple semi-supervised method—self-training—to further boost performance by utilizing the unlabeled portion of each target NER dataset. After the supervised fine-tuning step, we use the model to predict labels on each unannotated sentence in the target dataset, hereon referred to as the unsupervised set. For each sentence in the unsupervised set, we measure the average prediction confidence across all tokens within that sentence. Sentences whose average confidence meets a defined confidence threshold are then moved from the unsupervised set and added to the training set, using the predicted pseudo-labels as the ground truth labels.

We then repeat the supervised fine-tuning step by initializing a new model using the weights obtained from the pre-training step and then training on the enlarged training set (original training set plus high confidence sentences from the unsupervised set); however, we note that we retain the original validation set to ensure that only gold-standard labels are used for validation. Once the model has been trained, we once again apply self-training, predicting on the unsupervised set and moving high-confidence sentences into the train set. We repeat this process until no more sentences in the unsupervised set meet the required confidence threshold.

In our experiments, we set the confidence threshold to 99.75% average confidence across all tokens in a sentence to move that sentence from the unsupervised set to the train set; we discuss the choice of optimal confidence threshold in our Discussion section. For any given sentence, we obtain the average confidence from the BiLSTM-CRF by calculating the log-likelihood of the sequence of predicted labels (using the forward pass of the CRF) and then dividing by the number of words in the sentence. To obtain the average confidence for a given sentence from BERT, we collect the softmax score of each predicted label in the sequence and then average the scores.

## NER datasets

To evaluate the effectiveness of our methodology, we test the performance of our approach on eight commonly used biomedical NER datasets that cover different types of biomedical entities—BC2GM, BC4CHEMD, BC5CDR-chem, BC5CDR-disease, JNLPBA, NCBI-disease, Linnaeus, and S800. For all datasets, each data sample is composed of a sentence with word-level tokens $X = (w_1, \ldots, w_n)$ and associated word-level BIO annotations $Y = (y_1, \ldots, y_n)$. We note that while the datasets cover different entity types, within each dataset, entities are not annotated for type. Table 2 shows detailed descriptions of each dataset and how we split them into

**Table 2. Detailed information about each of our NER datasets.**

| Name | Entity Types | Train + Unsuperv. Sentences | Train + Unsuperv. Entities | Train + Unsupervised Entity Words/Total Words | Test Sentences | Test Entities | Test Entity Words/ Total Words |
|------|------|------|------|------|------|------|------|
| BC2GM | Gene/ Protein | 15093 | 18257 | .1050 | 5038 | 6325 | .1053 |
| BC4CHEMD | Drug/ Chem | 61321 | 58964 | .0728 | 26364 | 25346 | .0716 |
| BC5CDR-chem | Drug/ Chem | 9141 | 10550 | .0603 | 4797 | 5385 | .0563 |
| BC5CDR-disease | Disease | 9141 | 8427 | .0597 | 4797 | 4424 | .0574 |
| JNLPBA | Gene/ Protein | 18546 | 40753 | .2181 | 3856 | 6241 | .1647 |
| NCBI-disease | Disease | 6347 | 5921 | .0822 | 940 | 960 | .0836 |
| Linnaeus | Species | 16013 | 2824 | .0116 | 7142 | 1431 | .0136 |
| S800 | Species | 6563 | 2939 | .0398 | 1630 | 766 | .0428 |

**Table 3. Percent of unique entities in our NER datasets that also appear at least once as labeled entities in each of our pre-training datasets.**

|  | BC2GM | BC4 CHEMD | BC5CDR chem | BC5CDR disease | JNLPBA | NCBI disease | Linneaus | S800 |
|---|---|---|---|---|---|---|---|---|
| **100K SemMed** | 9.20% | 9.55% | 24.81% | 53.90% | 18.38% | 45.40% | 31.52% | 13.57% |
| **1M SemMed** | 12.45% | 17.20% | 36.53% | 67.67% | 23.35% | 54.50% | 41.25% | 22.21% |
| **MedMentions** | 10.48% | 10.42% | 26.77% | 52.21% | 26.38% | 44.72% | 33.07% | 16.45% |

train, unsupervised, and test sets. Each dataset is publicly available and can be downloaded from https://github.com/dmis-lab/biobert.

These eight NER datasets cover common biomedical entity types that are often extracted for various applications. Thus, these entity types are generally included within the UMLS metathesarus, and as some of the entities in these NER datasets may be covered within the pre-training datasets. In Table 3, we measure the percentage of unique entities in each NER dataset (train, dev, and test) that also appear at least once as labeled entities in the pre-training datasets. In our experiments, we evaluate the relationship between the amount of entity overlap and the effectiveness of transfer learning.

## Evaluation metrics

We adapt the entity-level evaluation metrics from the SemEval 2013 task 9.1 [66, 67]. For each task, we measure both exact precision, recall, and F1 as well as partial precision, recall, and F1. The exact metrics give credit only if the NER model correctly predicts the exact word boundaries for a given entity, while the partial metrics give partial credit if a NER model manages to predict part of an entity. Because our datasets are not annotated for entity types, we do not incorporate entity type into our evaluation. The calculations for each metric are described in Eqs 1–4:

$$\text{Possible} = \text{Correct} + \text{Incorrect} + \text{Partial} + \text{Missing} \tag{1}$$

$$\text{Actual} = \text{Correct} + \text{Incorrect} + \text{Partial} + \text{Spurious} \tag{2}$$

$$\text{Exact Precision} = \frac{\text{Correct}}{\text{Actual}}$$
$$\text{Exact Recall} = \frac{\text{Correct}}{\text{Possible}} \tag{3}$$
$$\text{Exact F1} = \frac{2 * \text{Exact Precision} * \text{Exact Recall}}{\text{Exact Precision} + \text{Exact Recall}}$$

$$\text{Partial Precision} = \frac{\text{Correct} + 0.5 \times \text{Partial}}{\text{Actual}}$$
$$\text{Partial Recall} = \frac{\text{Correct} + 0.5 \times \text{Partial}}{\text{Possible}} \tag{4}$$
$$\text{Partial F1} = \frac{2 * \text{Partial Precision} * \text{Partial Recall}}{\text{Partial Precision} + \text{Partial Recall}}$$

In the equations above, "Correct" refers to entities where the predicted boundaries exactly match the ground truth boundaries, "Partial" refers to entities where the predicted boundaries overlap but do not exactly match with the ground truth boundaries, "Missing" refers to entities that are in the ground truth labels but missed by the NER model, and "Spurious" refers to

entities predicted by the NER model but not actually in the ground truth labels. We note that "Incorrect" is used for incorrect entity types and is not applicable in our datasets.

# Results

## Comparing different pre-training corpora

A critical part of transfer learning is selecting an appropriate corpus on which to pre-train our models. In our study, we consider three different corpora to use for pre-training—∼100K random sentences from SemMed, ∼1M random sentences from SemMed, and the full MedMentions dataset.

In our first set of experiments, we evaluate the benefits of pre-training in settings where no labeled training data is available in the target domain. In Table 4 (see S1 Table for partial metrics), we show the performance of the BiLSTM-CRF and BlueBERT when pre-trained on each of the three different pre-training corpora and then directly applied to each of our target NER datasets without any fine-tuning. We also include the performance of MetaMap (2018 version) and scispaCy as performance baselines, as neither of these two popular NER tools requires fine-tuning for use. Our results show that when no fine-tuning on the downstream dataset is used, it is difficult to distinguish the effectiveness of pre-training on different corpora; in many cases, the popular MetaMap and scispaCy tools have comparable or better precision and recall than our pre-trained models. The ambiguity of these results suggests that if no labeled training samples are available in the downstream target dataset, there is no guarantee that pre-training a custom model for biomedical NER will work any better than simply using MetaMap or scispaCy.

**Table 4. Exact precision, recall, and F1 score of the BiLSTM-CRF and BlueBERT on each of our target datasets when pre-trained on different corpora without fine-tuning.**

| | BC2GM | BC4 CHEMD | BC5CDR chem | BC5CDR disease | JNLPBA | NCBI disease | Linneaus | S800 |
|---|---|---|---|---|---|---|---|---|
| **MetaMap (2018)** | EP: .0672 | EP: .0896 | EP: .1725 | EP: .1230 | EP: .0731 | EP: .0924 | EP: .0370 | EP: .0464 |
| | ER: .2903 | ER: .5045 | ER: .7257 | ER: .6299 | ER: .2743 | ER: .4731 | ER: .6457 | ER: .4720 |
| | EF: .1092 | EF: .1522 | EF: .2788 | EF: .2058 | EF: .1154 | EF: .1546 | EF: .0701 | EF: .0844 |
| **scispaCy** | EP: .0729 | EP: .0627 | EP: .1088 | EP: .0814 | EP: .0630 | EP: .0787 | EP: .0221 | EP: .0287 |
| | ER: .5025 | EP: **.6045** | EP: **.7853** | EP: .7141 | EP: .3792 | EP: .6090 | EP: **.6806** | EP: .5039 |
| | EF: .1273 | EP: .1137 | EP: .1911 | EP: .1461 | EP: .1081 | EP: .1395 | EP: .1429 | EP: .0544 |
| **BiLSTM-CRF** Pretrain 100K SemMed No FineTune | EF: .0780 | EP: .1078 | EP: **.1960** | EP: .1553 | EP: .0841 | EP: .1108 | EP: .0440 | EP: .0427 |
| | EF: .2496 | EP: .4549 | EP: .7015 | EP: .6770 | EP: .2221 | EP: .4604 | EP: .5500 | EP: .3081 |
| | EF: .1189 | EP: .1744 | EP: **.3063** | EP: .2527 | EP: .1220 | EP: .1786 | EP: .0815 | EP: .0750 |
| **BiLSTM-CRF** Pretrain 1M SemMed No FineTune | EF: .0761 | EP: .1044 | EP: .1930 | EP: .1503 | EP: .0970 | EP: .1108 | EP: .0426 | EP: .0494 |
| | EF: .2553 | EP: .4523 | EP: .7015 | EP: .6674 | EP: .2718 | EP: .4881 | EP: .5584 | EP: .3799 |
| | EF: .1172 | EP: .1697 | EP: .3030 | EP: .2454 | EP: .1430 | EP: .1806 | EP: .0792 | EP: .0875 |
| **BiLSTM-CRF** Pretrain MedMentions No FineTune | EF: **.1107** | EP: .0709 | EP: .1182 | EP: .0970 | EP: **.1448** | EP: .1026 | EP: .0271 | EP: .0355 |
| | EF: **.6114** | EP: .5525 | EP: .7288 | EP: .7267 | EP: **.6589** | EP: **.6616** | EP: .6730 | EP: **.5290** |
| | EF: **.1874** | EP: .1256 | EP: .2034 | EP: .1711 | EP: **.2375** | EP: .1776 | EP: .0521 | EP: .0666 |
| **BiLSTM-CRF** Pretrain 100K SemMed No FineTune | EF: .0659 | EP: **.1101** | EP: .1920 | EP: **.1703** | EP: .0920 | EP: .1302 | EP: **.0499** | EP: .0496 |
| | EF: .2022 | EP: .4504 | EP: .7099 | EP: **.7659** | EP: .2552 | EP: .5336 | EP: .5996 | EP: .3486 |
| | EF: .0994 | EP: **.1770** | EP: .3022 | EP: **.2787** | EP: .1353 | EP: .2094 | EP: **.0921** | EP: .0868 |
| **BiLSTM-CRF** Pretrain 1M SemMed No FineTune | EF: .0688 | EP: .1017 | EP: .1653 | EP: .1603 | EP: .1059 | EP: .1201 | EP: .0452 | EP: .0530 |
| | EF: .2167 | EP: .4029 | EP: .5585 | EP: .6586 | EP: .2774 | EP: .4901 | EP: .5625 | EP: .3893 |
| | EF: .1045 | EP: .1624 | EP: .2551 | EP: .2579 | EP: .1533 | EP: .1929 | EP: .0836 | EP: .0934 |
| **BiLSTM-CRF** Pretrain MedMentions No FineTune | EF: .0667 | EP: .1053 | EP: .1753 | EP: .1678 | EP: .1143 | EP: **.1324** | EP: .0444 | EP: **.0545** |
| | EF: .2109 | EP: .4292 | EP: .6144 | EP: .7150 | EP: .3125 | EP: .5551 | EP: .5535 | EP: .3956 |
| | EF: .1013 | EP: .1692 | EP: .2727 | EP: .2718 | EP: .1674 | EP: **.2138** | EP: .0822 | EP: **.0959** |

**Table 5. Exact precision, recall, and F1 score of the BiLSTM-CRF and BlueBERT on each of our target datasets when pre-trained on different corpora and fine-tuning on 1000 labeled sentences (800 train, 200 validation).**

| | BC2GM | BC4 CHEMD | BC5CDR chem | BC5CDR disease | JNLPBA | NCBI disease | Linneaus | S800 |
|---|---|---|---|---|---|---|---|---|
| **BiLSTM-CRF** | EP: .3087 | EP: .4181 | EP: .7427 | EP: .4267 | EP: .4711 | EP: .5699 | EP: .2480 | EP: .0089 |
| No Pretrain | ER: .2408 | ER: .4246 | ER: .7271 | ER: .4532 | ER: .5300 | ER: .5501 | ER: .0447 | ER: .0253 |
| FineTune 1000 | EF: .2706 | EF: .4213 | EF: .7349 | EF: .4395 | EF: .4989 | EF: .5598 | EF: .0758 | EF: .0132 |
| **BiLSTM-CRF** | EP: .3739 | EP: .5037 | EP: .7458 | EP: .6844 | EP: .5667 | EP: .6923 | EP: .6424 | EP: .6018 |
| Pretrain MedMentions | ER: .4527 | ER: .4417 | ER: .7889 | ER: .6784 | ER: .6860 | ER: .6821 | ER: .2199 | ER: .3667 |
| FineTune 1000 | EF: .4095 | EF: .4707 | EF: .7667 | EF: .6814 | EF: .6207 | EF: .6872 | EF: .3277 | EF: .4557 |
| **BiLSTM-CRF** | EP: .5740 | EP: .6344 | EP: .8098 | EP: .6704 | EP: .5662 | EP: .7006 | **EP: .6759** | EP: .6046 |
| Pretrain 100K SemMed | ER: .5488 | ER: .5469 | ER: .7916 | ER: .7057 | ER: .6675 | ER: .6292 | ER: .2874 | ER: .4200 |
| FineTune 1000 | EF: .5611 | EF: .5874 | EF: .8006 | EF: .6876 | EF: .6127 | EF: .6630 | **EF: .4033** | EF: .4957 |
| **BiLSTM-CRF** | **EP: .6584** | **EP: .6731** | **EP: .8415** | **EP: .7304** | **EP: .5914** | **EP: .7804** | EP: .4801 | **EP: .6697** |
| Pretrain 1M SemMed | **ER: .6157** | **ER: .5860** | **ER: .8332** | **ER: .7265** | **ER: .7046** | **ER: .7566** | **ER: .3270** | **ER: .4907** |
| FineTune 1000 | **EF: .6363** | **EF: .6265** | **EF: .8373** | **EF: .7284** | **EF: .6432** | **EF: .7683** | EF: .3890 | **EF: .5664** |
| **BlueBERT Base** | EP: .6178 | EP: .7244 | **EP: .8404** | **EP: .7448** | EP: .5502 | EP: .7645 | EP: .7446 | EP: .5185 |
| No Pretrain | ER: .6925 | ER: .6509 | ER: .8493 | ER: .7783 | ER: .7470 | ER: .8126 | ER: .3850 | ER: .4915 |
| FineTune 1000 | EF: .6531 | EF: .6857 | **EF: .8449** | **EF: .7612** | EF: .6337 | EF: .7878 | EF: .5076 | EF: .5047 |
| **BlueBERT Base** | **EP: .6480** | EP: .6745 | EP: .7788 | EP: .6802 | EP: .5958 | EP: .7738 | EP: .6325 | **EP: .6207** |
| Pretrain MedMentions | **ER: .7274** | **ER: .7087** | ER: .8779 | ER: .7929 | **ER: .7795** | **ER: .8435** | **ER: .6737** | ER: .4753 |
| FineTune 1000 | **EF: .6854** | **EF: .6912** | EF: .8254 | EF: .7323 | **EF: .6754** | **EF: .8071** | **EF: .6525** | EF: .5383 |
| **BlueBERT Base** | EP: .6439 | EP: .7593 | EP: .8332 | EP: .6939 | EP: .5939 | **EP: .7953** | EP: .8151 | EP: .5531 |
| Pretrain 100K SemMed | ER: .7044 | ER: .6249 | ER: .8380 | **ER: .8053** | ER: .7469 | ER: .8152 | ER: .4682 | ER: .6099 |
| FineTune 1000 | EF: .6728 | EF: .6856 | EF: .8356 | EF: .7454 | EF: .6617 | EF: .8051 | EF: .5948 | **EF: .5801** |
| **BlueBERT Base** | EP: .6338 | **EP: .7689** | EP: .7637 | EP: .7019 | **EP: .6067** | EP: .7430 | EP: .8143 | EP: .5265 |
| Pretrain 1M SemMed | ER: .7005 | ER: .6031 | **ER: .8906** | ER: .8015 | ER: .7400 | ER: .8235 | ER: .4780 | **ER: .6190** |
| FineTune 1000 | EF: .6655 | EF: .6760 | EF: .8223 | EF: .7484 | EF: .6668 | EF: .7812 | EF: .6024 | EF: .5690 |

In our next set of experiments, we include limited labeled training data from the target domain and then re-evaluate the effects pre-training. In Table 5 (see S2 Table for partial metrics), we show the performance of the BiLSTM-CRF and BlueBERT when pre-trained on each of the three different pre-training corpora and then fine-tuned on 1000 labeled sentences (800 train, 200 validation) from each target dataset. We also include the performance of the BiLSTM-CRF and BlueBERT directly trained on 1000 labeled sentences without any pre-training for comparison. In these experiments, the benefit of transfer learning becomes much clearer—all methods perform much better than the scispaCy and MetaMap baselines. We see that for the BiLSTM-CRF, pre-training on any of the three datasets results in better performance in both precision and recall than no pre-training. Of the three different pre-training corpora, pre-training on 1M sentences from SemMed gives the best overall performance. This suggests that for the BiLSTM-CRF model (and other similar models utilizing Word2Vec embeddings), it is most beneficial to pre-train on very large corpora, as the model is exposed to more useful vocabulary patterns and NER information.

On the other hand, for BlueBERT, the results between the different pre-training corpora are more mixed, and in some cases the base BlueBERT model (without any further pre-training) yields the strongest results. We expect that this is because the base BlueBERT model is already pre-trained using the masked-language-modeling task on all of Pubmed, so it may have already learned information useful for downstream NER. Our results show that further pre-training an existing BERT model such as BlueBERT on a large NER dataset is helpful for

some but not all downstream NER tasks. Pre-training BlueBERT on MedMentions resulted in the highest overall performance across the most downstream NER datasets. This may be because MedMentions, while smaller than the two SemMed corpora, is hand-labeled by humans and thus the labels are far more accurate; with an extremely powerful model such as BlueBERT that can learn extremely nuanced and subtle patterns, the quality of the labels may be more important than the quantity.

When we compare the overall improvement from pre-training with the entity overlap between the pre-training datasets and the target NER datasets shown in Table 3, we observe no clear relationship. For example, BC5CDR-disease has the largest overlap with the pre-training datasets. However, when comparing the improvement from pre-training using the BiLSTM-CRF, the magnitude of improvement is similar to that in BC2GM, which has the smallest amount of overlap. Furthermore, none of the pre-training datasets improved performance on BC5CDR-disease for BlueBERT. As another example, S800 has a very low overlap with the pre-training datasets, yet the magnitude of improvement from pre-training is far larger than in other datasets with more overlap. This indicates that low entity overlap in the pre-training dataset does not necessarily mean that transfer learning will not give a significant performance boost, and vice versa.

## Effects of transfer and semi-supervised learning

In Table 6 (see S3 and S4 Tables for partial metrics), we show the effects of transfer learning and semi-supervised learning on various NER datasets given different amounts of labeled training data. For all BiLSTM-CRF experiments, we pre-train the model on 1M sentences because it gave the overall strongest performance in Table 5. Likewise, for BlueBERT, we pre-train on MedMentions because it gave the overall strongest performance in Table 5. For both models, we also include the performance of a fully supervised version (trained on all available sentences in the train and unsupervised sets of each dataset, see Table 2 for the size of each dataset) without any pre-training for comparison.

When examining our BiLSTM-CRF results, we see that in general, more labeled data results in better performance in both transfer learning and semi-supervised learning. Compared to transfer learning without the self-training, the self-training step almost always provides an additional boost to performance; this performance boost is especially noticeable when there are few labeled training samples. In five of our eight NER datasets, combining transfer learning with self-training using 2000 labeled sentences (approximately 12%-30% of the total available labeled data depending on the dataset) yields similar or better performance than a fully supervised model trained on the full dataset.

We observe similar trends in our BlueBERT results. Increasing the amount of labeled data also increases the performance of both transfer and semi-supervised learning. Incorporating self-training on the unlabeled data provides a boost in F1 score on all but one dataset and training size (the only exception being BC4CHEMD with 250 labeled sentences); this difference is especially noticeable when the amount of labeled data is small. In five of our eight NER datasets, combining transfer learning with self-training using 2000 labeled sentences yields within 0.03 F1 score of fine-tuning BlueBERT on the full dataset. As expected, given the same training and data conditions, BlueBERT obtains notably better performance scores than the BiLSTM-CRF.

Our results show that in biomedical NER settings with small amounts of labeled training data, combining transfer learning and semi-supervised learning can boost precision and recall for both simple NER models such as a word-level BiLSTM-CRF and for more complex, state-of-the-art NER models such as BERT. We note that our experiments focus on downstream

**Table 6. Exact precision, recall, and F1 score of the BiLSTM-CRF and BlueBERT on each of our target datasets when fine-tuning on different amounts of labeled sentences, with and without semi-supervised self-training.** A fully supervised version is included for comparison. For all sets of training data, 80% of the available data is used for training and 20% of the available data is used for validation.

| | BC2GM | BC4 CHEMD | BC5CDR chem | BC5CDR disease | JNLPBA | NCBI disease | Linneaus | S800 |
|---|---|---|---|---|---|---|---|---|
| **BiLSTM-CRF** | EP: .4699 | EP: .4890 | EP: .7281 | EP: .5929 | EP: .4851 | EP: .6678 | EP: .0556 | EP: .2255 |
| Pretrain 1M SemMed | ER: .3493 | ER: .3887 | ER: .7615 | ER: .6654 | ER: .6315 | ER: .6072 | ER: .4655 | ER: .1816 |
| FineTune 250 | EF: .4007 | EF: .4331 | EF: .7444 | EF: .6270 | EF: .5487 | EF: .6361 | EF: .0994 | EF: .2012 |
| **BiLSTM-CRF** | EP: .6452 | EP: .6481 | EP: .7624 | EP: .7497 | EP: .5973 | EP: .7053 | EP: .1768 | **EP: .7749** |
| Pretrain 1M SemMed | ER: .4612 | ER: .4357 | ER: .8107 | ER: .5032 | ER: .6485 | ER: .6234 | ER: .5814 | ER: .0975 |
| FineTune 250 + SelfTrain | EF: .5379 | EF: .5211 | EF: .7858 | EF: .6022 | EF: .6219 | EF: .6618 | EF: .2712 | EF: .1724 |
| **BiLSTM-CRF** | EP: .4946 | EP: .6258 | EP: .8091 | EP: .6564 | EP: .5462 | EP: .7314 | EP: .4608 | EP: .6183 |
| Pretrain 1M SemMed | ER: .4162 | ER: .5584 | ER: .7915 | ER: .6890 | ER: .6743 | ER: .6751 | ER: .3358 | ER: .4309 |
| FineTune 500 | EF: .4520 | EF: .5902 | EF: .8002 | EF: .6723 | EF: .6035 | EF: .7021 | EF: .3885 | EF: .5078 |
| **BiLSTM-CRF** | EP: .6159 | EP: .6513 | EP: .8161 | EP: .7076 | EP: .6017 | EP: .7581 | EP: .2578 | EP: .7562 |
| Pretrain 1M SemMed | ER: .5155 | ER: .6019 | ER: .8185 | ER: .6680 | ER: .6870 | ER: .6902 | ER: .5902 | ER: .4088 |
| FineTune 500 + SelfTrain | EF: .5612 | EF: .6256 | EF: .8173 | EF: .6872 | EF: .6415 | EF: .7226 | EF: .3589 | EF: .5307 |
| **BiLSTM-CRF** | EP: .6584 | EP: .6731 | EP: .8415 | EP: .7304 | EP: .5914 | EP: .7804 | EP: .4801 | EP: .6697 |
| Pretrain 1M SemMed | ER: .6157 | ER: .5860 | ER: .8332 | ER: .7265 | ER: .7049 | ER: .7566 | ER: .3270 | ER: .4907 |
| FineTune 1000 | EF: .6363 | EF: .6265 | EF: .8373 | EF: .7284 | EF: .6432 | EF: .7683 | EF: .3890 | EF: .5664 |
| **BiLSTM-CRF** | EP: .6879 | EP: .7649 | EP: .8520 | EP: .7454 | EP: .6083 | EP: .7919 | EP: .7928 | EP: .7386 |
| Pretrain 1M SemMed | ER: .6691 | ER: .6140 | ER: .8466 | ER: .7267 | ER: .7133 | ER: .7819 | ER: .2889 | ER: .4328 |
| FineTune 1000 + SelfTrain | EF: .6784 | EF: .6812 | EF: .8493 | EF: .7359 | EF: .6566 | EF: .7869 | EF: .4234 | EF: .5458 |
| **BiLSTM-CRF** | EP: .6666 | EP: .7216 | EP: .8746 | EP: .7650 | EP: .6210 | **EP: .8130** | EP: .7285 | EP: .6785 |
| Pretrain 1M SemMed | ER: .6552 | ER: .6280 | ER: .8612 | ER: .7750 | ER: .7193 | ER: .8019 | ER: .4897 | ER: .6596 |
| FineTune 2000 | EF: .6608 | EF: .6716 | EF: .8678 | EF: .7700 | EF: .6665 | EF: .8074 | EF: .5857 | EF: .6689 |
| **BiLSTM-CRF** | **EP: .7208** | EP: .7766 | EP: .8810 | **EP: .7779** | EP: .6311 | EP: .8117 | **EP: .8653** | EP: .7251 |
| Pretrain 1M SemMed | **ER: .7173** | ER: .6629 | ER: .8723 | ER: .7713 | ER: .7250 | **ER: .8040** | ER: .4663 | **ER: .6769** |
| FineTune 2000 + SelfTrain | **EF: .7190** | EF: .7153 | EF: .8766 | **EF: .7746** | EF: .6748 | **EF: .8078** | EF: .6060 | **EF: .7001** |
| **BiLSTM-CRF** | EP: .7039 | **EP: .8665** | **EP: .8926** | EP: .7609 | **EP: .6769** | EP: .7764 | EP: .7974 | EP: .5774 |
| No Pretrain | ER: .7068 | **ER: .8534** | **ER: .8833** | **ER: .7772** | **ER: .7586** | ER: .7723 | **ER: .6408** | ER: .6079 |
| Fully Supervised | EF: .7053 | **EF: .8599** | **EF: .8879** | EF: .7690 | **EF: .7154** | EF: .7743 | **EF: .7106** | EF: .5923 |
| **BlueBERT Base** | EP: .4946 | EP: .3698 | EP: .7042 | EP: .5829 | EP: .5057 | EP: .6322 | EP: .4690 | EP: .4368 |
| Pretrain MedMentions | ER: .6051 | ER: .5349 | ER: .8296 | ER: .6741 | ER: .6632 | ER: .7116 | ER: .5339 | ER: .2953 |
| FineTune 250 | EF: .5443 | EF: .4373 | EF: .7618 | EF: .6252 | EF: .5738 | EF: .6696 | EF: .4993 | EF: .3524 |
| **BlueBERT Base** | EP: .5969 | EP: .6897 | EP: .7929 | EP: .6937 | EP: .6065 | EP: .7752 | EP: .5728 | EP: .5941 |
| Pretrain MedMentions | ER: .6436 | ER: .3088 | ER: .8378 | ER: .6801 | ER: .7395 | ER: .7254 | ER: .4892 | ER: .3635 |
| FineTune 250 + SelfTrain | EF: .6194 | EF: .4266 | EF: .8147 | EF: .6868 | EF: .6664 | EF: .7495 | EF: .5277 | EF: .4510 |
| **BlueBERT Base** | EP: .5955 | EP: .6267 | EP: .7524 | EP: .6183 | EP: .5621 | EP: .7188 | EP: .5224 | EP: .5115 |
| Pretrain MedMentions | ER: .6762 | ER: .6461 | ER: .8386 | ER: .7982 | ER: .7431 | ER: .7762 | ER: .5290 | ER: .4654 |
| FineTune 500 | EF: .6333 | EF: .6362 | EF: .7932 | EF: .6968 | EF: .6400 | EF: .7464 | EF: .5257 | EF: .4874 |
| **BlueBERT Base** | EP: .6670 | EP: .7817 | EP: .8636 | EP: .7767 | EP: .6578 | EP: .8042 | EP: .7442 | EP: .6822 |
| Pretrain MedMentions | ER: .7086 | ER: .6265 | ER: .8554 | ER: .7877 | ER: .7841 | ER: .7892 | ER: .4717 | ER: .4565 |
| FineTune 500 + SelfTrain | EF: .6872 | EF: .6955 | EF: .8595 | EF: .7822 | EF: .7154 | EF: .7966 | EF: .5774 | EF: .5470 |
| **BlueBERT Base** | EP: .6407 | EP: .6612 | EP: .8237 | EP: .6935 | EP: .5955 | EP: .7664 | EP: .6757 | EP: .5942 |
| Pretrain MedMentions | ER: .7040 | ER: .7160 | ER: .8755 | ER: .8290 | ER: .7798 | ER: .8466 | ER: .6436 | ER: .4831 |
| FineTune 1000 | EF: .6709 | EF: .6875 | EF: .8488 | EF: .7552 | EF: .6753 | EF: .8045 | EF: .6593 | EF: .5330 |
| **BlueBERT Base** | EP: .7381 | EP: .7702 | EP: .8495 | EP: .7623 | EP: .6462 | EP: .8426 | EP: .8091 | **EP: .6894** |
| Pretrain MedMentions | ER: .7205 | ER: .7021 | ER: .8866 | ER: .8201 | ER: .8061 | ER: .8637 | ER: .6932 | ER: .4746 |
| FineTune 1000 + SelfTrain | EF: .7292 | EF: .7346 | EF: .8676 | EF: .7901 | EF: .7174 | EF: .8530 | EF: .7467 | EF: .5622 |
| **BlueBERT Base** | EP: .7097 | EP: .7245 | EP: .8985 | EP: .8114 | EP: .6439 | EP: .8044 | EP: .7225 | EP: .6412 |

(*Continued*)

**Table 6.** (Continued)

| | BC2GM | BC4 CHEMD | BC5CDR chem | BC5CDR disease | JNLPBA | NCBI disease | Linneaus | S800 |
|---|---|---|---|---|---|---|---|---|
| Pretrain MedMentions | ER: .7345 | ER: .7540 | ER: .8694 | ER: .7856 | ER: .7663 | ER: .8703 | ER: .6296 | ER: .6554 |
| FineTune 2000 | EF: .7219 | EF: .7389 | EF: .8837 | EF: .7983 | EF: .6998 | EF: .8361 | EF: .6729 | EF: .6482 |
| **BlueBERT Base** | EP: .7613 | EP: .7854 | EP: .8903 | EP: .8040 | EP: .6755 | EP: .8460 | EP: .8745 | EP: .6887 |
| Pretrain MedMentions | ER: .7524 | ER: .7524 | ER: .9114 | ER: .8308 | **ER: .8144** | ER: .8583 | ER: .5891 | ER: .6797 |
| FineTune 2000 + SelfTrain | EF: .7568 | EF: .7686 | EF: .9007 | EF: .8172 | EF: .7385 | EF: .8521 | EF: .7040 | EF: .6841 |
| **BlueBERT Base** | **EP: .7940** | **EP: .8765** | **EP: .9113** | **EP: .8325** | **EP: .6932** | **EP: .8534** | **EP: .9136** | EP: .6756 |
| No Pretrain | **ER: .8175** | **ER: .8912** | **ER: .9248** | **ER: .8481** | ER: .8015 | **ER: .8755** | **ER: .7904** | **ER: .7249** |
| Fully Supervised | **EF: .8056** | **EF: .8838** | **EF: .9180** | **EF: .8402** | **EF: .7434** | **EF: .8643** | **EF: .8475** | **EF: .6994** |

NER applications with common biomedical entity types that overlap with the UMLS entity types covered in the pre-training datasets; we explore the effectiveness of these methods on a low-resource dataset with rare entity types in our Discussion section.

## Training time

We measured the approximate training times for each phase of our training methodology to give potential users a rough estimate of the associated computation requirements. All time measurements were performed using a single Tesla V100 GPU. For the BiLSTM-CRF, the pre-training step takes approximately one day for SemMed 1M; the fine-tuning step usually takes less than five minutes when using 1000 labeled sentences; and the semi-supervised step takes approximately one hour for the smallest dataset (NCBI-disease) to approximately sixteen hours for the largest dataset (BC4CHEMD). For BlueBERT, the pre-training step takes approximately one hour for MedMentions; the fine-tuning step usually takes less than ten minutes when using 1000 labeled sentences, and the semi-supervised step takes approximately three hours for the smallest dataset (NCBI-disease) to approximately two days for the largest dataset (BC4CHEMD).

## Application on low-resource datasets

One potential limitation of our study is that our pre-training datasets—SemMed and Med-Mentions—are labeled for UMLS entities and therefore may cover some of the target entities in our downstream test datasets. Thus, it is unclear how well transfer learning by pre-training on SemMed or MedMentions will help on downstream biomedical NER tasks where the target entity types are not covered by UMLS. To further explore this, we evaluate the effect of transfer learning and self-training using the 2018 Text Analysis Conference Systematic Review Information Extraction task (TAC SRIE) [68].

The TAC SRIE dataset (available online at https://tac.nist.gov/2018/SRIE/data.html) consists of the "Material and Methods" section from 100 scientific articles covering experiments where animals were exposed to environmental toxins and other environmental factors. Each text section is annotated by human toxicology experts for words and entities that describe the experimental design of the study; these include exposure (variable being tested, vehicle of delivery, purity of exposure, verification of exposure), animal group (control group, sample size, species, sex), dose group (dose amount, dose unit, dose frequency, dose duration, dose duration units, time of first dose, time of last time, time units), and endpoints (effect of dose, unit of measurement, time of measurement). We refer readers to [68] for more details about the entity types and the dataset. We note that the entity types annotated in TAC SRIE are

**Table 7. Detailed information about the TAC SRIE dataset.**

|  | Num Articles | Num Sentences | Num Entities | Entity Words/Total Words |
|---|---|---|---|---|
| **Labeled (All)** | 100 | 7993 | 15265 | .1607 |
| **Labeled (No Species/Sex)** | 100 | 7993 | 13029 | .1501 |
| **Unlabeled** | 344 | 31115 | n/a | n/a |

generally not within the entity types covered by UMLS and thus are likely to appear under different contexts than the entities from our pre-training datasets. The TAC SRIE dataset also includes "Material and Methods" sections from 344 additional articles that do not include any annotations. These articles are intended for evaluation, but the labels are not publicly available. For our experiment, we utilize these 344 articles as our unlabeled set for self-training.

For our evaluation, we utilize two versions of the TAC SRIE dataset. In the first version we include all annotations and entity types provided in the dataset. In the second version, we exclude annotations from the "species" and "sex" entity types; we exclude "species" because this entity type is most likely to overlap with UMLS and therefore the pre-training sets, and we exclude "sex" because this entity type is usually a simple keyword search for "male" or "female". We provide a summary of our TAC SRIE datasets in Table 7. We use 80/10/10 splitting on the labeled set to create train/val/test sets, and we use the same experimental setup as our main experiments where we pre-train our models, then fine-tune on the labeled set, and finally apply self-training on the unlabeled set. We note that TAC SRIE includes fine-grained entity type labels for each named entity; however, for our evaluation we do not predict specific entity types and only predict BIO annotations for entity or non-entity.

Table 8 shows the performance of the BiLSTM-CRF and BlueBERT on the TAC SRIE datasets with and without pre-training and self-training. For the BiLSTM-CRF, we see that pre-

**Table 8. Exact precision, recall, and F1 score of the BiLSTM-CRF and BlueBERT on the TAC SRIE datasets.** We show the effect of including pre-training and including semi-supervised self-training.

|  | TAC SRIE | TAC SRIE |
|---|---|---|
|  | **All Labels** | **No Species/Sex** |
| **BiLSTM-CRF** | EP: .4629 | EP: .3779 |
| No Pretrain | ER: .4497 | ER: .3489 |
| FineTune | EF: .4562 | EF: .3628 |
| **BiLSTM-CRF** | EP: .5842 | **EP: .5327** |
| Pretrain 1M SemMed | **ER: .5282** | ER: .4458 |
| FineTune | EF: .5548 | EF: .4854 |
| **BiLSTM-CRF** | **EP: .6036** | EP: .5237 |
| Pretrain 1M SemMed | ER: .5254 | **ER: .4646** |
| FineTune + SelfTrain | **EF: .5618** | **EF: .4924** |
| **BlueBERT Base** | EP: .6228 | EP: .5797 |
| No Pretrain | ER: .6228 | ER: .5934 |
| FineTune | EF: .6228 | EF: .5864 |
| **BlueBERT Base** | EP: .6055 | EP: .5784 |
| Pretrain MedMentions | ER: .6327 | ER: .5910 |
| FineTune | EF: .6188 | EF: .5846 |
| **BlueBERT Base** | **EP: .6889** | **EP: .6459** |
| No Pretrain | **ER: .6428** | **ER: .6166** |
| FineTune + SelfTrain | **EF: .6650** | **EF: .6309** |

training on 1M sentences from SemMed provides a large boost in precision and recall for both the full dataset and the dataset without species and sex annotations. However, the gain from self-training is very small and inconsistent. We expect that this is because the initial model performance prior to self-training is not high enough that self-training will propagate more knowledge than errors—we explore this further in our Discussion section.

On the other hand, we see that pre-training on MedMentions is not particularly helpful for BlueBERT compared to the base BlueBERT. This is not particularly surprising; we showed in our previous experiments that since BlueBERT is already pre-trained using masked-language-modeling, further pre-training using an NER dataset such as MedMentions sometimes but not always provides an additional performance boost. Unlike with the BiLSTM-CRF, self-training gives a noticeable boost in performance for BlueBERT. We expect that this is because the initial model performance prior to self-training is strong enough such that self-training can propagate more knowledge than errors.

Our results suggest that pre-training on UMLS entities and then self-training can be beneficial for downstream biomedical NER tasks even if they do not focus specifically on common UMLS entities. However, a more detailed study using a wider variety of low-resource biomedical NER tasks may be needed to establish the full scope of the benefits and limitations of our proposed methods in the context of low-resource settings.

## Discussion

### Self-training failure analysis

Based on our results in Table 6, we observe a general trend that utilizing semi-supervised self-training improves the overall F1 scores of the models, especially when there is a small amount of labeled data. However, in rare cases such as the BiLSTM-CRF on S800 with 250 initial labeled sentences, the overall F1 score drops significantly; multiple repeat runs showed the same behavior. One possible explanation for this behavior is that self-training propagates both knowledge and errors—a model that is highly confident when it is wrong will propagate bad labels during the self-training phase, thereby harming the performance of the final model. Therefore, when the model has an initial low performance before the self-training phase, self-training may not be as effective.

To better understand this phenomenon, we show the performance of the BiLSTM-CRF after each iteration of self-training under three different scenarios—S800 with 250 initial labeled sentences, Linnaeus with 250 initial labeled sentences, and BC2GM with 1000 initial labeled sentences (Fig 2). Linnaeus and S800 with 250 initial samples were chosen because the BiLSTM-CRF had the lowest F1 scores on these two datasets prior to self-training. In the S800 scenario, the performance of the model during the course of self-training is highly volatile. We observe that precision has a noticeable increase over time, especially in the early iterations; however, recall, which is already low to begin with, decreases over time causing the overall F1 score to be highly variable across the different iterations. Self-training on Linnaeus does not show this same behavior; precision, recall, and F1 score all show an initial increase and then hold at a fairly steady level through the remainder of the self-training process. Lastly, the self-training progress on BC2GM is representative of the typical self-training progression that we observed in most of the scenarios in this study—there are small/moderate gains in precision, recall, and F1 score over the course of self-training with occasional volatility caused by the inherent stochasticity associated with training deep learning models.

A common practice in self-training and other forms of semi-supervised learning is to continually iterate the semi-supervised method until no more samples meet the confidence threshold or some similar stopping criteria is met. However, our analysis shows that this practice

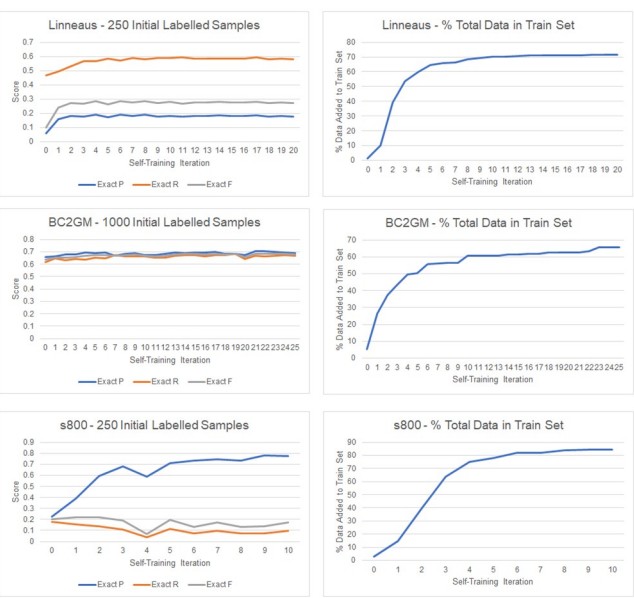

**Fig 2. Performance of the BiLSTM-CRF after each iteration of self-training for three scenarios: S800 with 250 initial labeled sentences, Linnaeus with 250 initial labeled sentences, and BC2GM with 1000 initial labeled sentences.** For each scenario, we also show the percentage of total available data (train + unsupervised) added to the training set after each iteration of self-training.

may not always yield higher performance, especially when the initial model has low performance. An alternative method, such as using validation score on a dedicated set of gold standard labels, may help safeguard against situations where self-training results in lower overall performance.

## Effect of transfer learning on self-training

As we have previously shown, semi-supervised learning can propagate both knowledge and errors; thus, semi-supervised approaches such as self-learning can be unreliable if the initial model has low performance. In settings with very few labeled examples, transfer learning can be critical in boosting initial model performance to levels where semi-supervised learning can provide a reliable boost. To demonstrate this effect, we analyzed the performance of self-training on the BC2GM dataset with and without transfer learning using both the BiLSTM-CRF and BlueBERT (Table 9).

For the BiLSTM-CRF, we observe that for all data sizes and all training scenarios, the pre-trained BiLSTM-CRF performs far better than the BiLSTM-CRF without pre-training. We note that when using the BiLSTM-CRF with no pre-training and 250 labeled sentences, no samples met the confidence threshold required to move data from the unsupervised set to the train set; therefore, self-training could not even be utilized. Compared to the pre-trained BiLSTM-CRF, the BiLSTM-CRF without pre-training also showed far greater instability in performance throughout self-training—performance often peaked in the early iterations of self-training and then slowly dropped in the later iterations. We observe a similar trend in the BlueBERT experiments in that for all data sizes and training scenarios, BlueBERT Base has lower F1 scores than BlueBERT Base with a second round of pre-training on MedMentions; however, the difference in performance is much smaller than in the BiLSTM-CRF. This is expected—as we showed in Table 5, BlueBERT Base already has strong performance in NER tasks because the base model is already pre-trained, and the second round of pre-training

**Table 9. Performance of the BiLSTM-CRF and BlueBERT on the BC2GM dataset with and without transfer learning.**

| | BiLSTM-CRF Pretrained 1M MetaMap | BiLSTM-CRF No Pretrain | BlueBERT Base + Pretrain MedMentions | BlueBERT Base |
|---|---|---|---|---|
| **BC2GM** Finetune 250 | EP: .4699 | EP: .0219 | EP: .4946 | EP: .4650 |
| | ER: .3493 | ER: .2424 | ER: .6051 | ER: .6424 |
| | EF: .4007 | EF: .0403 | EF: .5443 | EF: .5395 |
| **BC2GM** Finetune 250 + Self Train | EP: .6452 | EP: .0219 | EP: .5969 | EP: .5218 |
| | ER: .4612 | ER: .2424 | ER: .6436 | ER: .6536 |
| | EF: .5379 | EF: .0403 | EF: .6194 | EF: .5803 |
| **BC2GM** Finetune 500 | EP: .4946 | EP: .2423 | EP: .5955 | EP: .5711 |
| | ER: .4162 | ER: .1641 | ER: .6762 | ER: .6594 |
| | EF: .4520 | EF: .1957 | EF: .6333 | EF: .6121 |
| **BC2GM** Finetune 500 + Self Train | EP: .6159 | EP: .4114 | EP: .6670 | EP: .6334 |
| | ER: .5155 | ER: .2072 | ER: .7086 | ER: .6852 |
| | EF: .5612 | EF: .2756 | EF: .6872 | EF: .6583 |
| **BC2GM** Finetune 1000 | EP: .6584 | EP: .3376 | EP: .6407 | EP: .6178 |
| | ER: .6157 | ER: .3461 | ER: .7040 | ER: .6925 |
| | EF: .6363 | EF: .3418 | EF: .6709 | EF: .6531 |
| **BC2GM** Finetune 1000 + Self Train | EP: .6899 | EP: .4802 | EP: .7381 | EP: .7008 |
| | ER: .6691 | ER: .3643 | ER: .7205 | ER: .6977 |
| | EF: .6793 | EF: .4143 | EF: .7292 | EF: .6993 |
| **BC2GM** Finetune 2000 | EP: .6666 | EP: .4769 | EP: .7097 | EP: .6851 |
| | ER: .6552 | ER: .4680 | ER: .7345 | ER: .7536 |
| | EF: .6608 | EF: .4724 | EF: .7219 | EF: .7177 |
| **BC2GM** Finetune 2000 + Self Train | EP: .7208 | EP: .5042 | EP: .7613 | EP: .7324 |
| | ER: .7173 | ER: .5084 | ER: .7524 | ER: .7600 |
| | EF: .7190 | EF: .5063 | EF: .7568 | EF: .7460 |

using an NER-specific dataset is not guaranteed to always provide an additional performance boost.

These results show that transfer learning can be a critical tool in biomedical NER settings with very few labeled examples. When labeled data is extremely scarce, transfer learning may be required to bring the model up to a level of performance where semi-supervised learning can then be effectively applied. As shown in our experiments, the combination of transfer learning and semi-supervised learning can be a potent tool in improving performance in biomedical NER compared to a baseline model that uses neither, especially in situations where there are very few labeled sentences.

## Choosing the right confidence threshold for self-training

The selection of what confidence threshold to use for self-training can have a notable impact on the final performance of the NER model. For simplicity and consistency, we used 99.75% confidence as the threshold across all of our experiments—during the hyperparameter tuning phase, we observed that this confidence threshold returned generally strong results on most of the datasets. However, we note that this threshold is not guaranteed to be optimal under all settings.

In our experiments, we observed three general trends. (1) First, lower confidence thresholds require fewer iterations of self-training because each iteration adds more samples from the unlabeled set and therefore samples from the unlabeled set are used up more quickly. We noticed that for some datasets, extremely high thresholds also require fewer iterations of self-

**Table 10. We evaluate the BiLSTM-CRF pre-trained on SemMed1M on S800 with 250 labeled sentences and BC2GM with 1000 labeled sentences using different confidence thresholds for self-training.** We report exact precision, recall, and F1 score as well as the number of self-training iterations run before no more samples from the unlabeled set meet the confidence threshold.

| Confidence Threshold | S800 (250 labeled) Score | S800 (250 labeled) Iterations | BC2GM (1000 labeled) Score | BC2GM (1000 labeled) Iterations |
|---|---|---|---|---|
| 0.9 | EP: .4199 | | EP: .6331 | |
| | **ER: .1575** | 3 | ER: .6504 | 6 |
| | EF: .2291 | | EF: .6417 | |
| 0.95 | EP: .5022 | | EP: .6611 | |
| | ER: .1495 | 5 | ER: .6761 | 13 |
| | **EF: .2305** | | EF: .6685 | |
| 0.99 | EP: .6316 | | EP: .6818 | |
| | ER: .1122 | 8 | **ER: .6773** | 27 |
| | EF: .1905 | | **EF: .6795** | |
| 0.9975 | **EP: .7749** | | EP: .6899 | |
| | ER: .0975 | 10 | ER: .6691 | 25 |
| | EF: .1724 | | EF: .6793 | |
| 0.999 | EP: .6854 | | **EP: .6954** | |
| | ER: .0697 | 18 | ER: .6468 | 18 |
| | EF: .1265 | | EF: .6702 | |

training because after a number of initial iterations, no more samples from the unlabeled set make it pass the threshold. (2) Second, too low or too high of a confidence threshold results in lower performance in terms of overall F-score; the optimal range for the confidence threshold varies by dataset. Finally, (3) the specific behavior of how different confidence thresholds affect precision, recall, and F-score is dependent on the dataset and model.

In Table 10, we show how different confidence thresholds affect self-training using the BiLSTM-CRF (pre-trained on SemMed 1M) on S800 with 250 labeled sentences and on BC2GM with 1000 labeled sentences. On S800, we observe that lower thresholds improve recall at the expense of precision, whereas higher thresholds improve precision at the expense of recall. On BC2GM, this trend is much weaker, and we see that when the confidence threshold is set too low both precision and recall drop. In both datasets, setting the confidence threshold too high or too low causes the overall F-score to reduce; furthermore, the confidence threshold that produces the highest over F-score is not the same between the two datasets.

From these results, we see that it is difficult to define a universal "best" confidence threshold that will work well for all situations. Instead, users will likely need to tune the confidence threshold as a hyperparameter based on the needs of the specific application.

## Conclusion

In this work, we evaluated the effectiveness of combining transfer learning with semi-supervised learning to perform biomedical NER in applications with limited amounts of labeled training data and that focus on common biomedical entities such as those covered in UMLS. We used two different base models—a BiLSTM-CRF and BlueBERT—and evaluated on eight different standard biomedical NER datasets covering different types of common biomedical entities. For each dataset, we generated scenarios with different amounts of available labeled data—250, 500, 1000, and 2000 labeled sentences.

For each model, we first evaluated the effect of pre-training on three different corpora—∼ 100K sentences from SemMed, ∼1M sentences from SemMed, and all ∼50K sentences from MedMentions. We found that for the BiLSTM-CRF model, pre-training on 1M sentences

from SemMed provided the largest boost in performance. Since BlueBERT is already pre-trained, the effect of the second round of pre-training was less consistent. Overall, further pre-training of BlueBERT on MedMentions gave the best results.

Next, we evaluated the effect of incorporating semi-supervised self-training into each model. For both the BiLSTM-CRF and BlueBERT, we found that in almost all scenarios, self-training gave a boost to the final F1 scores; this boost was especially large in scenarios with very few labeled sentences (250 and 500 initial labeled sentences). Because self-training can propagate both knowledge and errors, in rare cases where the model performance was very low before self-training was applied, self-training had inconsistent results and sometimes lowered the F1 score. In our analysis, we showed that transfer learning is critical in scenarios with very few labeled sentences to bring the model performance up to levels where self-training can be effective.

One limitation of our study is that our experiments focused on downstream NER tasks with common entity types that are covered by UMLS. As a result, the UMLS entities annotated in our pre-training datasets may overlap with the entities in the downstream NER tasks. Therefore, it is unclear how much pre-training and self-training will help in downstream NER tasks that utilize entity types not covered in UMLS. To help address this limitation, we showed that pre-training and self-training can still boost performance when applied to TAC SRIE, a low-resource dataset where the goal is to extract entities from toxicology papers that are related to experimental procedures; the entity types of interest in TAC SRIE are generally not covered within the entity types from UMLS. However, we note that a broader study utilizing a wider range of different types of low-resource NER datasets is required to establish the effectiveness of our methods in low-resource settings.

In this work, we utilized self-training for our semi-supervised method, which is an extremely simple method. We expect that more sophisticated semi-supervised methods, such as co-training or tri-training using models pre-trained on different corpora, may provide better performance. Future work also includes evaluating the effect of transfer learning and semi-supervised learning on datasets where predicting entity type is part of the NER task. The code used for our experiments is available online at https://code.ornl.gov/biomedner/biomedner.

## Supporting information

**S1 Table. Exact and partial precision, recall, and F1 score of the BiLSTM-CRF and Blue-BERT on each of our target datasets when pretrained on different corpora without fine tuning.**
(TIF)

**S2 Table. Exact and partial precision, recall, and F1 score of the BiLSTM-CRF and Blue-BERT on each of our target datasets when pretrained on different corpora and fine tuning on 1000 labeled samples (800 train, 200 validation).**
(TIF)

**S3 Table. Exact and partial precision, recall, and F1 score of the BiLSTM-CRF on each of our target datasets when fine tuning on different amounts of labeled data, with and without semi-supervised self-training. A fully supervised version is included for comparison.**
For all sets of training data, 80% of the available data is used for training and 20% of the available data is used for validation.
(TIF)

**S4 Table. Exact and partial precision, recall, and F1 score of BlueBERT on each of our target datasets when fine tuning on different amounts of labeled data, with and without semi-**

**supervised self-training. A fully supervised version is included for comparison.** For all sets of training data, 80% of the available data is used for training and 20% of the available data is used for validation.
(TIF)

## Author Contributions

**Conceptualization:** Shang Gao, Olivera Kotevska, Alexandre Sorokine.

**Data curation:** Shang Gao, Olivera Kotevska, Alexandre Sorokine.

**Formal analysis:** Shang Gao, Olivera Kotevska, Alexandre Sorokine.

**Funding acquisition:** J. Blair Christian.

**Investigation:** Shang Gao, Olivera Kotevska, Alexandre Sorokine.

**Methodology:** Shang Gao, Olivera Kotevska, Alexandre Sorokine.

**Project administration:** J. Blair Christian.

**Software:** Shang Gao.

**Supervision:** J. Blair Christian.

**Visualization:** Shang Gao.

**Writing – original draft:** Shang Gao, Olivera Kotevska.

**Writing – review & editing:** Shang Gao, Olivera Kotevska, Alexandre Sorokine.

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
