## [Decision Letter · Decision Letter 0]

21 Oct 2020

PONE-D-20-29224

Biomedical named entity recognition in low resource settings

PLOS ONE

Dear Dr. Gao,

Thank you for submitting your manuscript to PLOS ONE. After careful consideration, we feel that it has merit but does not fully meet PLOS ONE’s publication criteria as it currently stands. Therefore, we invite you to submit a revised version of the manuscript that addresses the points raised during the review process.

Particularly, the reviewers highlighted a bias in the experiment setup, which does not simulate low-resource settings properly. There are a few other important remarks, especially with regards to code availability as PLOS One dedicates great efforts for data availability. Finally, the manuscript would benefit from a few grammar/typo corrections. 

We look forward to receiving your revised manuscript.

Kind regards,

Nicolas Fiorini

Academic Editor

PLOS ONE

Journal Requirements:

2. Thank you for providing links to the data used in the experiment in your Data Availability Statement. We ask that you also include these links within your Methods section.

4. Please include your tables as part of your main manuscript and remove the individual files. Please note that supplementary tables (should remain/ be uploaded) as separate "supporting information" files.

Reviewers' comments:

Reviewer's Responses to Questions

**Comments to the Author**

1. Is the manuscript technically sound, and do the data support the conclusions?

Reviewer #1: Partly

Reviewer #2: Partly

2. Has the statistical analysis been performed appropriately and rigorously? 

Reviewer #1: N/A

Reviewer #2: Yes

3. Have the authors made all data underlying the findings in their manuscript fully available?

Reviewer #1: Yes

Reviewer #2: Yes

4. Is the manuscript presented in an intelligible fashion and written in standard English?

Reviewer #1: No

Reviewer #2: Yes

5. Review Comments to the Author

Reviewer #1: This paper investigated cumulative effect of pre-training and semi-supervised self-training on multiple datasets. The pre-training was executed on three datasets, SemMed 100k, SemMed 1M, and MedMentions. The authors showed that the self-training method with the pre-trained parameters can boost the performance. Although the performance improvement by pre-training and semi-supervised self-training was previously known in the NER task, this paper has contribution on the cumulative effect of pre-training and semi-supervised self-training in low resource settings. However, this paper constructed a low resource setting by reducing the number of training samples from a large size of corpus. Thus, it is important to evaluate the proposed approach to the real application with a small number of NER annotation.

Major:

1. The proposed approach was tested on the applications with a large number of NER datasets such as disease names, chemicals, and genes. Because UMLS semantic types, which were used for pre-training, include entities such as chemicals and phenomenon or process, pre-training datasets already included many similar entities. The authors need to present the domain similarity between the tested dataset and pre-training datasets to make sure that the experiment settings are real low resource settings.

2. The authors mentioned that CRISPR and Covid-19 are examples of the application areas with a low resource. Thus, it is important to test the proposed approach using a real low resource data set, where the size of a corpus set is small and similar pre-training data sets are not available.

3. What was the criterion to set the confidence threshold to 99.75%? When set to lower threshold, the model is likely to propagate errors more than the knowledge. How can we find the proper threshold?

Minor

- Please check the grammar errors. For examples, "We note the this", "NER ragging", "naturaly language processing" and so on.

Reviewer #2: I would like to thank the authors for this well-written and easy to read article. The article proposes that transfer learning and self-training can be beneficial for biomedical NER tasks with small training sets. Through various ablation studies they show the benefit of transfer learning and self-training individually and also delve into some of the unexpected results in the Discussion. I really like the paper, but unfortunately don’t feel that the paper’s conclusions on “low-resource settings” are warranted. While I accept that pretraining and self-training improve performance, there needs to be better comparisons to show it in context with other methods. I describe my concerns below.

Major

- I don’t accept that the datasets used for testing and the methods for downsampling them are actually low-resource settings. The entity data used for pretraining is all based on UMLS terms which are a superset of all the downstream datasets evaluated. A truly low resource setting would be one with only a small number of relevant annotations for the task in hand. These can often be found in languages other than English or for more obscure biomedical NER tasks, compared to the comparatively mainstream tasks examined. Further to that, the concept of transfer learning involves training on one different task first and then training on the target task. The prior task in this work is predicting UMLS terms, which is a superset of the different biomedical entities in the eventual target tasks. The results in Table 3 shows this, as the classifiers can predict many of the entities (even with poor F1) with no fine-tuning on the task. So again, this transfer learning primarily works because the prior task is so similar to the final target task. This is an interesting result, but should be framed appropriately.

- Comparing against SemRep (MetaMap) and Scispacy aren’t great baselines. Both of them seem to be predicting UMLS terms so haven’t been trained specifically for the tasks. For a fuller context, we really need better comparisons. You either need to show how other good NER tools behave with such small training sets, or use your methods for one of the smaller tasks with the complete dataset + pretraining + self-training and compare it to other tools.

- The statement about code availability is not reasonable and code should be made available during peer review

- Removing statements about low-resource settings, the main conclusions seem to be that transfer learning from UMLS and self-training often gives a boost for biomedical NER, where the entities are a subset of UMLS and in short supply. This is a good contribution and is well-written but the scope and conclusions need to be reframed accordingly.

Minor

- The mentions of CRISPR & COVID as potential low resource settings seem click-baity. It also doesn’t relate to any of the datasets used and should be removed, or be explained and better supported.

- The paper makes reference to the SemRep tool for entity extraction. I believe they really mean MetaMap, which is the underlying entity extraction tool. SemRep extracts relations between these entities.

- For the results presented in Table 5, we really need to know what fraction of the complete dataset is being used for Fine Tuning. This would enable clearer comparison with the Fully Supervised set. Is 2000 samples a lot of the dataset for each one or a fraction?

- In Figure 2, “active learning iteration” is a confusing axis label? Perhaps “Iteration of self-training”?

- The paper is very well-written and easy to follow. There are a number of tiny mistakes listed below:

- “we generate pseudo-labels” -> “we generate psuedo-labels”

- “a wide range of naturaly language processing” -> “a wide range of natural language processing”

- “showed that pretraining an BiLSTM-CRF” -> “showed that pretraining a BiLSTM-CRF”

- “labels for NER ragging” -> “labels for NER tagging”

- “SciScpacy” -> “scispaCy”

- “both transer and semi-supervised” -> “both transfer and semi-supervised”

6. PLOS authors have the option to publish the peer review history of their article (what does this mean?). If published, this will include your full peer review and any attached files.

Reviewer #1: No

Reviewer #2: No

---

## [Author Response · Author response to Decision Letter 0]

19 Nov 2020

Please see reviewer response file.

---

## [Decision Letter · Decision Letter 1]

14 Dec 2020

PONE-D-20-29224R1

A pre-training and self-training approach for biomedical named entity recognition

PLOS ONE

Dear Dr. Gao,

Thank you for submitting your manuscript to PLOS ONE. After careful consideration, we feel that it has merit but does not fully meet PLOS ONE’s publication criteria as it currently stands. Therefore, we invite you to submit a revised version of the manuscript that addresses the points raised during the review process.

First of all, thank you for having revised your manuscript in such depth. This has been appreciated by the reviewers who now believe the manuscript is a lot more suitable for publication. There are a few remaining minor remarks that would be nice to be addressed though. Thanks again for your work and contribution to PLOS ONE.

We look forward to receiving your revised manuscript.

Kind regards,

Nicolas Fiorini

Academic Editor

PLOS ONE

Reviewers' comments:

Reviewer's Responses to Questions

**Comments to the Author**

1. If the authors have adequately addressed your comments raised in a previous round of review and you feel that this manuscript is now acceptable for publication, you may indicate that here to bypass the “Comments to the Author” section, enter your conflict of interest statement in the “Confidential to Editor” section, and submit your "Accept" recommendation.

Reviewer #1: (No Response)

Reviewer #2: (No Response)

2. Is the manuscript technically sound, and do the data support the conclusions?

Reviewer #1: Yes

Reviewer #2: Yes

3. Has the statistical analysis been performed appropriately and rigorously? 

Reviewer #1: N/A

Reviewer #2: Yes

4. Have the authors made all data underlying the findings in their manuscript fully available?

Reviewer #1: Yes

Reviewer #2: Yes

5. Is the manuscript presented in an intelligible fashion and written in standard English?

Reviewer #1: Yes

Reviewer #2: Yes

6. Review Comments to the Author

Reviewer #1: In the first review, I had four comments. In this revision, the reviewers were addressed those comments.

Reviewer #2: I would like to thank the authors for their substantial revisions and responses to our comments. I feel the paper has been dramatically improved and that the problem reframing seems a lot more suitable. I have a few remaining issues and a few minor things that I hope can be addressed. Overall, I think this is excellent work and has made me think a lot about self-training and methods of transfer learning.

- At the moment, the first result (Table 4) reads like an odd experiment. It seems that the hypotheses is that pretraining alone might provide okay performance for problems with no training samples. Could you move that justification (which seems to be in the final sentence of the paragraph discussing Table 4) further up so it's clear why you're showing those results. At first read, I couldn't remember why you were showing results of a model without any fine-tuning which feels weird and had to stop and read around a bit.

- For the MetaMap/scispacy baselines, do you filter for entities of the relevant type for each dataset? They are presumably predicting all UMLS terms and I believe should normalize them to UMLS entities, which could be filtered by type. As an example, do you only include MetaMap predictions of UMLS terms that are of type "gene/protein" in the BC2GM dataset?

- I wonder why you've put the TAC SRIE result in the Discussion section. It adds nice weight to your argument about NER outside of UMLS. It seems like it would be a nice final result, but really that's up to you.

And here are a few small tweaks.

- In related work, I'm not sure I would describe a BiLSTM-CRF as simple. I get that it is comparitively simple compared to a language model like BERT. But there are much simpler NER frameworks, so that probably needs a rephrase.

- Page 15, 18, 21 & Table 5: "Medmentions" -> "MedMentions"

- Page 22, 23, Table 8 caption & Table 10: "Semmed" -> "SemMed"

7. PLOS authors have the option to publish the peer review history of their article (what does this mean?). If published, this will include your full peer review and any attached files.

Reviewer #1: No

Reviewer #2: No

---

## [Author Response · Author response to Decision Letter 1]

8 Jan 2021

Please see attached cover letter and reviewer response document.

---

## [Editor Report · Decision Letter 2]

18 Jan 2021

A pre-training and self-training approach for biomedical named entity recognition

PONE-D-20-29224R2

Dear Dr. Gao,

We’re pleased to inform you that your manuscript has been judged scientifically suitable for publication and will be formally accepted for publication once it meets all outstanding technical requirements.

Kind regards,

Nicolas Fiorini

Academic Editor

PLOS ONE

Additional Editor Comments (optional):

Thank you for substantially improving the manuscript and thoroughly responding to reviewers throughout the process.

---

## [Editor Report · Acceptance letter]

29 Jan 2021

PONE-D-20-29224R2 

A pre-training and self-training approach for biomedical named entity recognition 

Dear Dr. Gao:

I'm pleased to inform you that your manuscript has been deemed suitable for publication in PLOS ONE. Congratulations! Your manuscript is now with our production department. 

Kind regards, 

on behalf of

Dr. Nicolas Fiorini 

Academic Editor

PLOS ONE